# Overcoming the Negligence in Laboratory Diagnosis of Mucosal Leishmaniasis

**DOI:** 10.3390/pathogens10091116

**Published:** 2021-09-01

**Authors:** Lilian Motta Cantanhêde, Cristiane Batista Mattos, Ana Karoline Cruz, Yoda Janaina Ikenohuchi, Flavia Gonçalves Fernandes, Enmanuella Helga Ratier Terceiro Medeiros, Cipriano Ferreira da Silva-Júnior, Elisa Cupolillo, Gabriel Eduardo Melim Ferreira, Ricardo de Godoi Mattos Ferreira

**Affiliations:** 1Laboratory of Genetic Epidemiology, FIOCRUZ, Porto Velho 76812245, Rondonia, Brazil; cristianebmattos@gmail.com (C.B.M.); anakarolinechruz@gmail.com (A.K.C.); yodajanaina@gmail.com (Y.J.I.); fernandesflaviag@gmail.com (F.G.F.); ehrtttm@gmail.com (E.H.R.T.M.); drciprianoferreira@hotmail.com (C.F.d.S.-J.); gabriel.ferreira@fiocruz.br (G.E.M.F.); ricardo.godoi@fiocruz.br (R.d.G.M.F.); 2Leishmaniasis Research Laboratory, Oswaldo Cruz Institute, FIOCRUZ, Rio de Janeiro 21040360, Rio de Janeiro, Brazil; elisa.cupolillo@ioc.fiocruz.br

**Keywords:** tegumentary leishmaniasis, mucosal leishmaniasis, molecular diagnosis

## Abstract

The northern region of Brazil, which has the largest number of cases of tegumentary leishmaniasis (TL) in the country, is also the region that has the highest diversity of species of vectors and *Leishmania* parasites. In this region, cases of mucosal leishmaniasis (ML), a clinical form of TL, exceed the national average of cases, reaching up to 12% of the total annual TL notifications. ML is associated with multiple factors, such as the parasite species and the viral endosymbiont *Leishmania* RNA virus 1 (LRV1). Being a chronic parasitological disease, laboratory diagnosis of ML poses a challenge for health services. Here, we evaluated more than 700 clinical samples from patients with clinical suspicion of TL, including patients with cutaneous leishmaniasis (CL) and mucosal leishmaniasis, comparing the results of parasitological tests—direct parasitological examination by microscopy (DP) and conventional PCR (cPCR) targeting of both kDNA and hsp70. The DP was performed by collecting material from lesions through biopsies (mucosal lesions) or scarification (cutaneous lesions); for PCR, a cervical brush was used for sample collection. Blood samples were tested employing standardized real-time PCR (qPCR) protocol targeting the HSP70 gene. PCR tests showed higher sensitivity than DP for both CL and ML samples. Considering ML samples only (N = 89), DP showed a sensitivity of 49.4% (N = 44) against 98.8% (N = 88) for kDNA PCR. The qPCR hsp70 for blood samples from patients with ML (N = 14) resulted in superior sensitivity (50%; N = 7) compared to DP (21.4%; N = 3) for samples from the same patients. Our results reinforced the need to implement a molecular test for the diagnosis of ML, in addition to proposing methods less invasive for collecting material from TL patients. Sample collection using a cervical brush in lesions observed in CL and ML patients is easy to perform and less invasive, compared to scarification and biopsies. Blood samples could be a good source for qPCR diagnosis for ML patients. Thus, we propose here a standardized method for collection and for performing of molecular diagnosis of clinical samples from suspicious ML patients that can be applied in reference services for improving ML diagnosis.

## 1. Introduction

Tegumentary leishmaniasis (TL) is an endemic disease, with an estimated 600,000 to 1 million new cases occurring globally each year, with the highest concentration in America, the Mediterranean basin, the Middle East, and Central Asia. In Brazil, it is endemic throughout the territory and Brazil was among the few countries responsible for about 90% of the new cases of TL in 2019 [1,2].

TL is present in two clinical forms—cutaneous leishmaniasis (CL; localized, disseminated, or diffuse) and mucosal leishmaniasis (ML; late or concomitant), among other rare variations [3,4,5]. In CL, the most common clinical form is the localized lesion. ML is a clinical evolution of the CL form where the late mucosal lesion appears after a CL event [6,7].

The parasite is transmitted to the vertebrate host by the bite of the infected female phlebotomine insect during the blood meal. Different species of *Leishmania* parasite can cause the infection. For workers involved in activities related to environmental changes, such as deforestation, construction of dams, and activities in the field, such as agriculture, mining, and farming, the chances of acquiring infection are increased [2].

One of the biggest challenges in controlling TL is the diversity of vectors and reservoirs involved in the *Leishmania* life cycle and the diversity of parasite species. The Amazon region has the greatest known diversity [8], with most of *Leishmania* species linked to human infections already described in this region [8,9,10,11,12].

In general, 3–5% of asymptomatic individuals or those with recovered CL develop ML [13,14,15]. In Brazil, the North Region contributes to the most significant number of TL cases, and the percentage of ML cases is higher than the national average. In some endemic municipalities, the proportion of ML cases compared to the total number of notified cases can reach 25% [16].

The factors involved in clinical evolution are diverse and include the immune response developed by the mammalian host, the virulence of the parasite, questions related to treatment, and, in some endemic areas, the presence of the endosymbiont *Leishmania* RNA virus 1 (LRV1) [6,17,18,19,20,21,22]. Factors linked to treatment, including absence, complete non-adherence, and refractoriness, are reported to be associated with ML cases [23,24,25]. 

The diagnosis of TL is based on clinical evidence, but *Leishmania* infection is confirmed by laboratory tests. In Brazil, 80% of TL cases are confirmed by laboratory diagnostics, but in other regions, such as Argentina, only 30% of TL cases are based on clinical and laboratory information [26].

Laboratory examinations for TL include a direct parasitological test (DP), preferably used in health services and polymerase chain reaction (PCR), as well as isolation and cultivation of the parasite in culture medium. The sensitivity of DP is considerably lower in samples of lesions with scarce amastigotes, a characteristic influencing ML diagnosis [27,28]. For this reason, DP is not recommended for ML diagnosis, which is based on the intradermal Montenegro test (IDRM), PCR, histopathology, or parasite in vitro isolation. Clinical examinations based on rhinoscopy or oropharyngoscopy are also useful for ML diagnosis. Thus, ML is a chronic lesion that is difficult to diagnose, due to the low parasitic load and the need for specialized professionals to collect the biological sample or perform the main examinations [29]. Among the options provided, PCR and conventional or quantitative methods are tests that have the greatest advantages in terms of sensitivity and specificity [30,31,32,33].

There are several PCR-based methods for diagnosis of TL, targeting kinetoplast DNA (kDNA), transcribed internal spacer (ITS), and heat shock protein 70 gene (*hsp*70), among others [31,34,35,36,37,38,39]. Each marker has both advantages and disadvantages, thus, validation with clinical samples is necessary.

Thus, considering the importance of determining a molecular marker that could be applied for clinical samples, the present study aimed to evaluate the diagnostic sensitivity of PCR directed towards two widely used molecular markers (kDNA and *hsp*70) in a large number of patients with cutaneous and mucosal lesions, proposing the use of a non-invasive method for collecting samples that provides an adequate amount of genetic material for performing molecular diagnosis tests. We also analyzed, in a smaller number of samples, the performance of a qPCR assay employing DNA from venous blood from patients with ML showing that this could be a good source for the diagnosis of these patients.

## 2. Results

We investigated 723 cases with clinical suspicion of tegumentary leishmaniasis (TL) and performed the three parasitological tests—a direct parasitological test (DP) and two PCR-based tests, kDNA and hsp70 PCR. Of these, 557 samples (77.04%) were positive in at least one of the three parasitological tests applied, with an overall agreement of 76.15% (fixed-marginal kappa = 0.50; 95% CI for fixed-marginal kappa (0.46, 0.55) and the two molecular tests together resulted in 75.6% positive diagnostic, showing an overall agreement of 81.63% (fixed-marginal kappa = 0.59; 95% CI for fixed-marginal kappa (0.51, 0.67). Considering the three parasitological tests employed, 41.21% (N = 298) of samples were positive to all. Fewer than 50% of clinically suspicious patients would have been diagnosed if only DP was employed (Table 1). One hundred twenty-three patients presented mucosal lesions and 72.4% were confirmed with *Leishmania* spp. infection, but only 36% were confirmed based on DP.

The following statistical analyses consider only positive cases (N= 557) for TL based on one of the three parasitological tests employed. In addition, we analyzed the performance of a qPCR protocol in 14 blood samples from patients with ML. These results will be shown after exploring the results of the 557 samples.

The analysis of clinical and epidemiological data shows that the average age of patients diagnosed with TL was 41.7 years and ranged from a minimum of 2 to a maximum of 86 years, with 42.9% (N = 239) of positive cases in the age range between 20 and 40 years old. Male gender was predominant in 438 cases (78.6%). Cutaneous lesions were more frequent than ML, totaling 468 (84.0%) cases, and in 470 (84.4%) of cases the acquisition of the infection reported by patients was in work situations. For 130 (23.3%) patients, presence of previous TL was reported and of these, 125 (95.5%) declared that they had been previously treated for infection (Table 2). It was not possible to assess whether the reported treatment was specific to TL or whether it was carried out as recommended by the Ministry of Health of Brazil (medication/time/dose). 

The number of lesions at the time of collection was assessed, ranging from one to five lesions. A total of 94% (N = 523) of patients with positive results had a single lesion.

Most of the positive samples were detected by at least one of the molecular tests (N = 547; 98.2%), but ten samples were positive for DP only and 53.5% were positive for the three tests (Table 3), but there was poor agreement among them (69% overall agreement; fixed-marginal kappa= 0.08). The lowest agreement was observed when comparing results obtained for ML samples by DP and molecular tests—48% and 54% related to PCR kDNA and PCR hsp70, respectively.

The sensitivity of tests applied varied according to the type of lesion. For samples of cutaneous lesions, the three tests (DP, kDNA, and *hsp*70) were positive in 56.2% (N = 263) of samples. In mucosal lesions, only 39.3% (N = 35) of samples presented positive results in the three tests (Table 3).

We evaluated each test separately. The kDNA test had the highest positivity, showing 528 (94.8%) positive samples, followed by *hsp*70 and DP, presenting 433 (77.7%) and 351 (63%) positive samples, respectively (Figure 1).

The difference in sensitivity by type of lesion was due to the low sensitivity of direct parasitological examination for samples from mucosal lesions, showing negative results for 45 (51%) samples, whereas negative DP result was obtained for 206 (37%) samples of cutaneous lesions (Figure 1).

To determine the sensitivity parameters of the applied tests, we considered the following as a clinical reference: (i) the set of tests, which included the three applied tests and (ii) the combined molecular tests, which were the samples with positive results in one of the molecular tests (kDNA, *hsp*70, or both).

When the clinical reference was the set of tests, the kDNA showed a sensitivity of 95%, while the DP showed 63% sensitivity. When the clinical reference was the combined molecular tests, the kDNA showed 97% sensitivity against 62% for the DP. The same inferences of sensitivity were made considering the type of lesion, cutaneous (CL), or mucosal (ML). In the results of the tests applied to the cutaneous lesions, we observed that the kDNA showed 99% sensitivity in the set of tests and 100% sensitivity when using the parameter of combined molecular tests. The DP showed a sensitivity of 66% in the set of tests and 65% in the combination of molecular tests. For mucosal lesions, in the set of tests, the test with the highest sensitivity was the kDNA at 94%, and the test with the lowest sensitivity was the DP at 49% (Table 4).

Positive samples based on PCR *hsp*70 (N = 433) were subjected to *Leishmania* species identification by RFLP. A total of 340 samples (84.8%) resulted as *L. braziliensis* and 43 samples (10.7%) as *L. guyanensis*. Four other *Leishmania* species were identified: *L. lainsoni* (N = 8; 1.85%), *L. shawi* (N = 7; 1.62%), *L. amazonensis* (N = 2; 0.42%), and *L. lindenbergi* (N = 1; 0.23%) (Table 5).

In 32 samples (7.39%), it was not possible to identify the species involved in the infection (Table 5), because they had fragments with a different profile from the *Leishmania* species known for the *hsp*70 marker or because they had poor intensity fragments, making it impossible to visualize on an agarose gel.

### The qPCR in Blood Samples from ML Patients

Whole-blood samples were collected from 14 (15.9%) ML patients with a positive result for kDNA PCR. DNA extracted from these samples was analyzed using a standardized qPCR *hsp*70 protocol [35]. The time of appearance of mucosal lesions was variable in these patients, ranging from recent lesions (0.25 months) to old lesions (72 months) (Table 6). The majority (N = 11) of these patients reported previously having cutaneous leishmaniasis, and all of them performed an initial treatment with Glucantime^®^. Two patients did not complete the initial treatment and one patient received a second treatment with amphotericin (Table 6).

Only three (21%) of these 14 patients presented positive results by DP using samples collected from mucosal lesions. The qPCR *hsp*70 using DNA from mucosal lesions was positive in 10 (71.4%) samples, while the same protocol in blood samples was positive in 7 (50%) samples. Only one positive sample in the DP was negative in the *hsp*70 qPCR using DNA from lesions, but it was positive when using DNA from blood (Patient 7). Two other samples were positive for blood and negative for lesion (patients 1 and 14). These results may be associated with DNA degradation since the qPCR *hsp*70 test for mucosal lesions samples was performed using DNA stocks which were employed for both kDNA and *hsp*70 conventional PCR.

## 3. Discussion

A total of 77% (N = 557) of the analyzed samples from lesions of patients with clinical suspicion of TL were positive for at least one of the applied parasitological tests. However, we cannot affirm that the patients with a negative result for all samples analyzed by the different tests performed did not present TL. The absence of a test considered the gold standard for diagnosing TL, or even a standardized clinical reference, impairs the estimate of the classic parameters of diagnostic accuracy of the tests (sensitivity, specificity, and positive and negative predictive value) [40].

Demographic and clinical variables (sex, age, type of lesion, and the form of disease acquisition) were used based on the 557 samples that tested positive for at least one of the three parasitological tests. The profile of patients presenting clinical and parasitological diagnosis for TL was similar to the observed profile in other areas of the Amazon region. For example, in two other states of this region, Acre and Amazonas, about 80% of TL cases are males, aged between 15 and 45 years [41,42,43]. The form of acquisition addressed was not performed according to the other works recorded in the literature, where the type of activity (agriculture, arms, or gold miner) performed by the patient was registered. Here, we only addressed whether the acquisition was made during work or leisure activities.

Almost a quarter of positive patients (N = 130; 23.3%) reported the existence of previous TL (43.08% of ML and 56.92% of CL), and of these, 95.5% reported having done some type of treatment, but it was not possible to determine if this was specific for LT and, if so, the protocol was the recommended by the Ministry of Health was followed [44]. One of the main factors related to the recurrence of TL is therapeutic failure, which is reported in up to 33% of cases, being associated with the species of the parasite [45,46], the presence of LRV1 [47,48], the host’s immune response [49], the type of lesion [13], and other factors.

Cutaneous lesions were the most frequent clinical form observed in the patients with a positive test result (84.02%), but compared to other Brazilian regions, a high number of ML cases were observed. However, it is important to mention that this study was carried out in a Reference Center, and it is likely that a considerable number of patients presenting recurrent clinical forms, such as the case of ML and even more complicated skin lesions, are referred to this center. Thus, it is not possible to infer that these numbers represent the actual percentages of CL and ML cases in the geographic area where the study was conducted. In general, the proportion of ML in South America is 3.1% for TL cases, however, this number varies according to the frequency of the species and the genetic and immunological aspects of the host, as well as the availability of diagnosis and treatment [50]. In Brazil, on average, 3 to 5% of patients evolve to ML [13,51]. In Bolivia, a country bordering the state of Rondônia, this percentage varies from 12 to 14.5% [52], similar to what was observed in this study. It is important to note that, aside any region in Brazil, the government does not support leishmaniasis treatment in Bolivia and patients must pay for it, which can result in metastatic lesions. In addition, the viral endosymbiont LRV1 exists in a portion of the parasites circulating in the Brazilian Amazon and in five other countries that comprise the International Amazon [22,47,48,53,54]. This may also imply the highest ML rates found in these regions since LRV1 was associated with evolution to a mucosal form [22]. Interestingly, in other regions of Brazil that have a lower incidence of LM, LRV1 was investigated and found at low or no frequency [55,56].

The control and management of TL in endemic regions such as the Amazon encounters numerous obstacles, and this could be associated with the high frequency of ML observed here. In addition to the diverse faunal composition, another factor that contributes negatively is the fact that endemic sites lack primary care services, mainly due to the low population density of these regions, which leads to delays in seeking medical attention or often, to treating the disease with alternative or not-recommended methods. The absence of treatment or incomplete treatment is related to cases of recurrence of the disease, including the worsening of the clinical prognosis and development of mucosal lesions [57,58,59].

The etiology of the disease is among the factors related to TL progression. A previous study conducted in Rondônia by our group has already described the scenario of *Leishmania* species causing CL [22], and the frequency of species was very similar to that observed in this study. *L. braziliensis*, the species more frequently associated with ML, was the most frequent species (84.79%) among 401 samples identified (Table 5), followed by *L. guyanensis*, a species responsible for about 70% of TL cases in Amazonas, the only state in Brazil where *L. braziliensis* is not the principal etiological agent of TL [60,61]. Other species (*L. lainsoni*, *L. shawi*, *L. amazonensis*, and *L. lindenbergi*) were detected causing cutaneous lesions in the patients included in this study, corroborating studies showing a diversity of *Leishmania* species associated with CL in the Amazon region [22,62,63]. Some regions bordering the state of Rondônia, such as Mato Grosso [64] and Bolivia [65], also present this pattern of high frequency of *L. braziliensis* causing human disease. Although, the same is observed in Acre, showing *L. braziliensis* responsible for about 60% of human cases of TL, *L. shawi* is the second most frequent species in this region [43].

Although, molecular tests are pointed out as an important tool for diagnosis of TL, DP is still the most widely used test; it is performed quickly and easily, in addition to having high specificity. However, this approach presents low sensitivity, ranging from 40 to 74.4%, especially when compared with the sensitivity of molecular tests. Sensitivity of this test varies according to the type of lesion, method of sample collection and preparation of slides, and the microscopist experience [39,66,67]. Although in this study, standardized protocol was employed for sample collection and slide preparation, and the same professional examined all the slides, 206 patients would not have been diagnosed if only DP was performed. Molecular tests are more sensitive for TL than for DP, with sensitivity for TL of up to 100% [35,68,69,70]. As expected, higher sensitivity was observed for molecular tests (PCR) employed in this work, and only 10 samples positive for DP were negative for both molecular tests. PCR targeting minicircle kDNA is considered the most sensitive molecular target for both CL and ML diagnosis, with reports of up to 97% sensitivity [71,72,73]. High sensitivity is also reported for PCR *hsp*70, although, this approach showed lower sensitivity than PCR kDNA [35,74,75]. Our results are in agreement with the sensitivity reported for the different tests. Lower sensitivity was observed for DP in mucosal lesions (49%) compared to the sensitivity in skin lesions (66%), reinforcing the influence of the sample analyzed and corroborating the results showing few parasites in mucosal lesions [76]. However, the sensitivity of both PCR approaches was about the same for samples from mucosal and cutaneous lesions, despite some reports showing differences in sensitivity of molecular markers by type of lesion [77]. The collection method used was the same for cutaneous and mucosal lesions. In addition, the samples were stored for a short period of up to 7 days, under refrigeration at –20 °C, until molecular analysis. These factors may have influenced the similar sensitivity observed between the types of lesions.

Despite the fact that we found different species in studied samples, it was not possible to infer whether the parasite species influenced the sensitivity of the tests due to the small number of samples identified with species other than *L. braziliensis*.

The three parasitological tests had different levels of sensitivity, with more significant variation for DP, showing a reduced sensitivity for mucosal leishmaniasis samples, compromising the use of this diagnostic test for ML. This finding shows that suspicious ML patients need to be better assisted. Mucosal lesions are considered a clinical evolution of CL and require a more prolonged treatment [44]. Early diagnosis can avoid further complications, with facial involvement. Mucosal lesions usually have a low parasitic burden and sample collection for laboratory diagnosis is not an easy task, which are both factors that influence the low sensitivity of DP. Other laboratory tests, in addition to DP, are employed for ML diagnosis, such as histopathological tests, but biopsy is needed for this, resulting in an invasive and painful collection, which can only be performed by a specialist [29]. Besides, the result of the histopathological examination, in the absence of parasites in lesions and with the finding of diffuse granulomatous dermatitis, can be confused with other diseases [78], such as histoplasmosis, caused by *Histoplasma capsulatum* [79,80], a disease that can be found in endemic regions for TL, such as the Amazon. Thus, the method of sample collection and the PCR approaches employed in this study for molecular detection of *Leishmania* parasites seems to be a good alternative for ML diagnosis.

The results of blood samples from patients with mucosal leishmaniasis indicate that the use of blood samples has potential for ML diagnosis. The collected samples are simple and do not cause damage to the affected patient, as a biopsy, which is commonly used does. Blood samples associated with molecular methods have already been widely used for visceral leishmaniasis diagnosis and showed sensitivity equal to or better than bone marrow or splenic aspirates [81,82]. For CL diagnosis, as expected, the blood samples even with leukocyte enrichment showed reduced sensitivity compared to samples obtained directly from the lesion [38,69], but they also showed that the dissemination of the parasite with the detection of DNA in the blood occurs even before the appearance of the mucosal lesion and is persistent after treatment [83,84,85]. Previous research has reported the isolation of the parasite in direct culture of blood samples from patients with ML [86,87,88]. The use of blood samples for the molecular diagnosis of ML has been poorly explored, but the results shows that it is more sensitive in comparison with CL patients [84]. Our results reinforce that the use of blood samples is promising for the diagnosis of ML and the discordant results in the analyses (Table 6) emphasize the need to use an internal control of the reaction, such as RNAseP, as in the original publication [35]. It is also necessary to validate or exclude possible methodological flaws. A study with a larger number of blood samples and with control of the reactions is necessary to assess the use of venous blood more accurately in the diagnosis of ML.

## 4. Material and Methods

### 4.1. Collection of Clinical Data and Biological Samples

A total of 723 samples were collected from patients with clinical suspicion of TL whose attended the Hospital of Tropical Medicine (CEMETRON), Rondônia, Brazil, from July 2013 to June 2017. Individual questionnaires were used to collect personal and demographic data and clinical information, including duration of symptoms and number and location of cutaneous lesions. Two biological samples were collected from each patient. For the direct parasitological (DP) examination, samples were collected by scarifying the inner edge of the lesion using scalpel blades and placed in slides. For DNA extraction, further employed in the polymerase chain reactions (PCR), a sterile cervical brush was used to collect samples from the edge of the cutaneous lesions or directly from the mucosal lesions (nasal or oral) and stored in 2 mL microtubes at –20 °C until DNA extraction.

In the final 8 months (November 2016 to June 2017) of the study, blood samples from all patients with ML were included in the analyses, totaling 14 samples. Blood samples were collected in tubes containing the anticoagulant, ethylenediaminetetraacetic acid (EDTA).

### 4.2. Examination of Samples

#### 4.2.1. Direct Parasitological (DP) Tests

Slides containing scarified material collected from the lesion were stained with Giemsa, as recommended by the Ministry of Health for Brazil [44]. For each sample, three imprints were made on the slide to increase the sensitivity of the test. The positive result was determined by the visualization of amastigote structures in the slides.

#### 4.2.2. kDNA and hsp70 PCR

DNA extraction was performed with PureLink DNA MiniKit (Invitrogen^®^, Carlsbad, CA, USA). The lysis reagents were added to the 1.5 mL microtube containing the cervical brush according to the kit protocol. The cervical brush was discarded in the step that preceded the transfer of the lysate to the silica column.

The amplification of the kinetoplast DNA (kDNA) was performed using the primers already described [89], which amplify a fragment of approximately 120 base pairs. The reactions were prepared in a final volume of 25 μL containing 4 μL of total DNA, 0.4 μM of each primer, 200 μM of dNTP, 0.75 mM MgCl2, and 1 U Taq DNA Polymerase Invitrogen (Life Technologies, Carlsbad, CA, USA). The test was performed in a thermocycler with the following protocol: 94 °C for 5 min followed by 35 cycles at 94 °C for 30 s, 57 °C for 45 s, 72 °C for 30 s, and finally, a final extension at 72 °C for 5 min. The PCR for hsp70 and species identification by RFLP were performed as previously described, with the amplification of a 234 bp fragment and subsequent restriction reaction with the enzymes *Hae*III and *BsTU*I [22,36]. As a positive control of molecular reactions, DNA extracted from parasites cultivated in culture medium (*L. braziliensis*—MHOM/BR/2015/285, *L. guyanensis*—MHOM/BR/2015/391, and *L. amazonensis*—IFLA/BR/1967/PH8) with PureLink DNA MiniKit (Invitrogen, Carlsbad, CA, USA), following the protocol of the kit for extraction of culture cell. After the PCR reaction, the amplified fragments were subjected to horizontal electrophoresis (200V for 30 min) on 2% agarose gel and stained with GelRed ™ (Biotium^®^, Hayward, CA, USA), and the images of the gels were obtained in a photo-documenter model Image Quant LAS4000 (GE Healthcare Bio-Sciences AB, Uppsala, Sweden).

#### 4.2.3. Real-Time PCR for Blood Mucosal Samples

Blood samples were collected in tubes containing EDTA, stored at 4 °C for up to 36 h, and then extracted. According to the manufacturer’s recommendations, from 200 µL of whole blood, extraction was performed with the Illustra blood genomic Prep Mini Spin Kit (Ge Healthcare, Chicago, IL, USA). In the final step, the DNA was eluted in 100 µL and stored at –20 °C until use. A clinical validated qPCR assay targeting HSP70 gene [35], combined with a SYBRGreen protocol, was employed for *Leishmania* DNA detection in blood samples from 14 selected ML patients, all presenting positive results for PCR kDNA. Reactions were performed in technical duplicates with a final volume of 25 μL containing 2 μL of total DNA, 0.3 μM of each primer, and 12.5 μL Power SYBR Green Master Mix (Applied Biosystems, Carlsbad, CA, USA). The tests were performed on the 7500 Real-Time PCR Systems equipment (Applied Biosystems, USA) under the following conditions: 50 °C for 2 min (activation of amperase) and 95 °C for 10 min (initial denaturation), followed by 40 cycles at 95 °C for 15 s (denaturation), 61 °C for 1 min (annealing and extension), and 60 °C for 1 min (dissociation curve).

#### 4.2.4. Statistical Analyses

A descriptive summary of the observed variables was presented using direct counts and percentages. The studied variables were summarized as discrete, nominal values. The accessed variables described in the above sections could be categorized into two groups, (1) general epidemiological aspects—gender, type of lesion, putative local of infection, previous leishmaniasis, and treatment in the first infection; and (2) *Leishmania* detection methods (DP, kDNA, and hsp70).

The absence of an appropriate gold-standard test and the impact of this absence is part of the article discussion, including the need of a definition of clinical reference to be considered through the analyses and discussions. The agreement between two methods was formally compared using classical statistic kappa, as implemented by Randolph [90].

## 5. Conclusions

Combination of molecular PCR-based approaches with a non-invasive collection technique are promising tools for the diagnosis of tegumentary leishmaniasis [91,92]. We demonstrated that molecular tests have important benefits for use in the diagnosis of mucosal leishmaniasis and should be applied as the first choice for the diagnosis of these lesions by the reference centers that have capacity and technologies available for this application. A sterile cervical brush is a good alternative method for collecting samples from mucosal lesions; with the advantage, compared to biopsy, of being a non-invasive method that can be performed by different health professionals, it brings benefits to the patient and the health system.

The results obtained from blood samples from patients with mucosal lesions confirm the potential of using molecular methods for the diagnosis of ML. Despite the small number of ML evaluated, the detection of parasite DNA in the blood was more sensitive than the direct parasitological test on the mucosal lesions.

## Figures and Tables

**Figure 1 pathogens-10-01116-f001:**
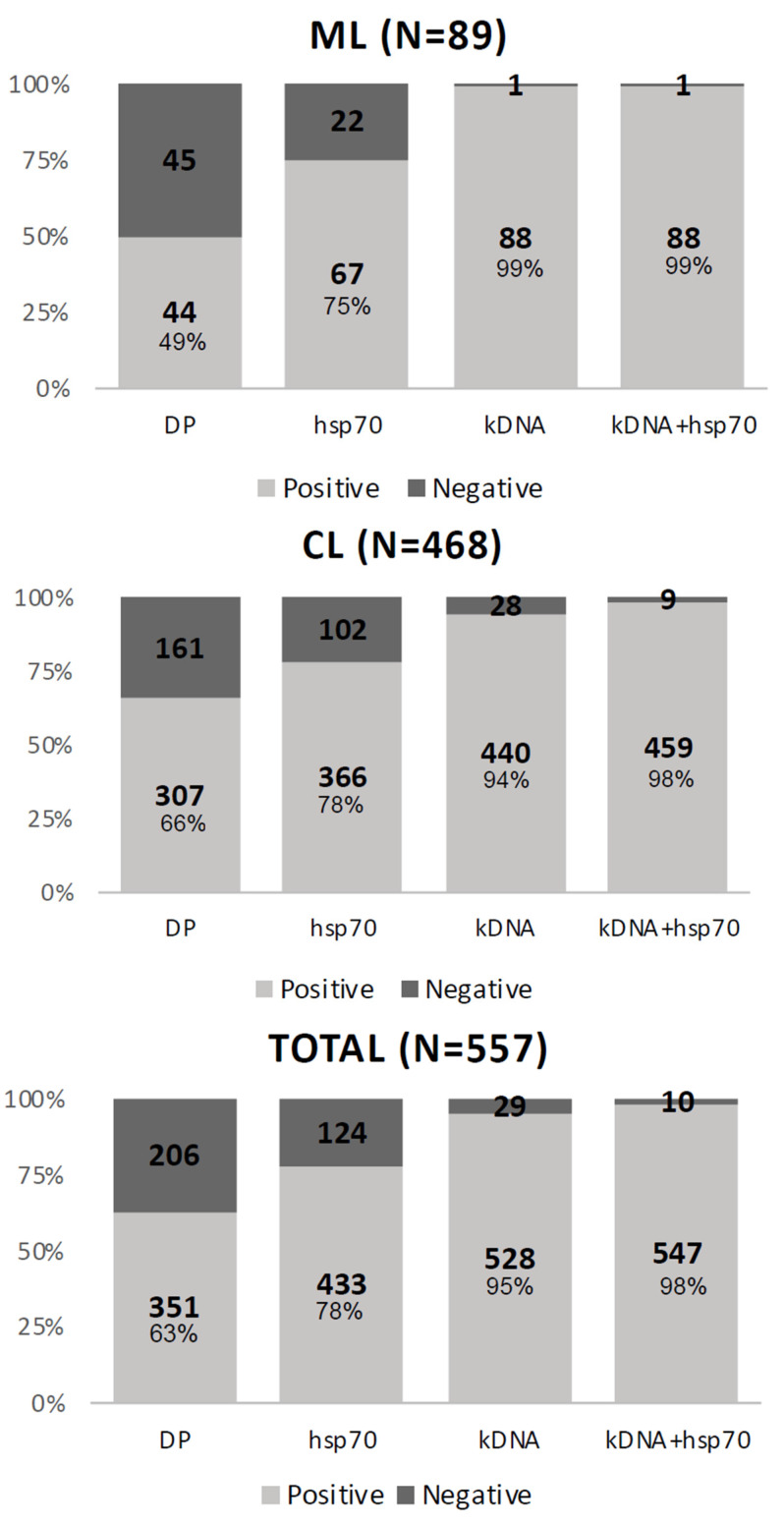
Results of parasitological tests applied to the total number of samples and by type of clinical manifestation, cutaneous lesion (CL) or mucosal lesion (ML).

**Table 1 pathogens-10-01116-t001:** Absolute frequencies based on the results of parasitological tests for 723 patients with clinical suspicious of tegumentary leishmaniasis.

Test	Result	CL + ML (%)	CL (%)	ML (%)
DP	Positive	351 (48.4)	307 (51.2)	44 (36.0)
	Negative	372 (51.6)	293 (48.8)	79 (64.0)
kDNA	Positive	528 (73.0)	440 (73.3)	88 (71.5)
	Negative	195 (27.0)	160 (26.7)	35 (28.5)
hsp70	Positive	433 (59.9)	366 (61.0)	67 (54.5)
	Negative	290 (40.1)	234 (39.0)	56 (45.5)
Positive for one of the molecular tests	Yes	547 (75.7)	459 (76.5)	88 (71.5)
	No	176 (24.3)	141 (23.5)	35 (28.5)
Positive for one of parasitological tests	Yes	557 (77.0)	468 (78.0)	89 (72.4)
	No	166 (23.0)	132 (22.0)	34 (27.6)

**Table 2 pathogens-10-01116-t002:** Absolute frequencies of data collected for 557 patients with clinical and laboratory diagnostic for leishmaniasis.

Variable	Categories	N	%
Gender	Male	438	78.6
	Female	119	21.4
Type of lesion	Cutaneous	468	84.0
	Mucosal	89	16.0
Putative local of infection	Work	470	84.4
	Leisure	87	15.6
Previous leishmaniasis	Yes	130	23.3
	No	427	76.7
Treatment in the first infection	Yes	5	4.5
	No	125	95.5
	Not applicable *	427	
DP	Positive	351	63.0
	Negative	206	37.0
kDNA	Positive	528	94.8
	Negative	29	5.2
hsp70	Positive	433	77.7
	Negative	124	22.3
Positive for one of the molecular tests	Yes	547	98.2
	No	10	1.8

* Only patients with previous TL or referred for recurrence were considered in this variable. N = number of samples.

**Table 3 pathogens-10-01116-t003:** Distribution of the results of direct parasitological tests (DP), PCR kDNA, and PCR *hsp*70 for positive cases of TL (CL + ML) and for cases separated into cutaneous lesions (CL) and mucosal lesions (ML).

Test			CL + ML		CL		ML	
DP	kDNA	*hsp*70	N	**%(+)**	N	%(+)	N	%(+)
+	+	+	298	53.5	263	56.2	35	39.3
+	+	-	40	7.2	32	6.8	8	9.0
+	-	+	3	0.5	3	0.6	0	0
+	-	-	10	1.8	9	1.9	1	1.1
-	+	+	116	20.8	84	17.9	32	36.0
-	+	-	74	13.3	62	13.2	13	14.6
-	-	+	16	2.9	16	3.4	0	0
Total			557		468		89	

N = number of samples. % (+) = Percentage of positive cases.

**Table 4 pathogens-10-01116-t004:** Inference of the sensitivity of the tests, considering, as a clinical reference, the set of tests (kDNA, *hsp*70, and DP) and the combined molecular tests (kDNA and *hsp*70) in the total of samples (CL + ML) and by type of cutaneous lesion (CL) or mucosal (ML).

			CL + ML			CL			ML		
“Clinical reference”	Test		“Clinical reference”		Sensibility	“Clinical reference”		Sensibility	“Clinical reference”		Sensibility
		Result	+	-		+	-		+	-	
Set of tests	DP	+	351	0	0.63	307	0	0.66	44	0	0.49
		-	206	166		161	132		45	34	
	kDNA	+	528	0	0.95	440	0	0.99	83	5	0.94
		-	29	166		28	132		1	34	
	*hsp*70	+	433	0	0.78	366	0	0.78	67	0	0.75
		-	124	166		102	132		22	34	
	Combined molecular	+	547	0	0.98	459	0	0.98	88	0	0.99
		-	10	166		9	132		1	34	
Combined molecular tests	DP	+	341	10	0.62	298	9	0.65	43	1	0.49
		-	206	166		161	132		45	34	
	kDNA	+	528	0	0.97	440	0	1	83	5	0.96
		-	19	176		19	141		0	35	
	*hsp*70	+	433	0	0.79	366	0	0.8	67	0	0.76
		-	114	176		93	141		21	35	

**Table 5 pathogens-10-01116-t005:** Frequency of *Leishmania* species in *hsp*70-positive samples.

Species	N	% (*hsp*70+)	% (ID+)
*L. braziliensis*	340	78.52	84.80
*L. guyanensis*	43	9.93	10.70
*L. lainsoni*	8	1.85	2.00
*L. shawi*	7	1.62	1.75
*L. amazonensis*	2	0.46	0.50
*L. lindenbergi*	1	0.23	0.25
Unidentified fragment pattern	32	7.39	-

*hsp*70+ = percentages relative to the set of *hsp*70-positive cases. ID+ = percentages relative to the set of cases in which it was possible to determine the species. N = number of samples.

**Table 6 pathogens-10-01116-t006:** Evaluation of qPCR hsp70 for parasitological diagnostic of mucosal leishmaniasis employing blood samples from patients with mucosal lesions. Fourteen patients were selected based on positive results for PCR kDNA from mucosal lesion samples; clinical and epidemiological information was collected during the patient’s attendance.

Patient Information				Parasitological Test			
				Lesion			Blood
Patient	Time of ML (months)	Previous leishmaniasis/Year	Treatment in the first infection	PCR kDNA	DP	qPCR hsp70	qPCR hsp70
1	72	Yes/1979	Glu/60 doses	+	-	-	+
2	7	Yes/1994	Incomplete treatment	+	+	+	+
3	6	No	--	+	-	+	+
4	24	Yes/1974	Glu/50 doses	+	-	+	-
5	6	Yes/2014	Glu/109 doses	+	+	+	-
6	60	Yes/ND	Incomplete treatment	+	-	-	-
7	24	Yes/2016	Glu/32 doses; Anfo/09 doses	+	+	-	+
8	0.25	Yes/2015	Glu/60 doses	+	-	+	+
9	2	No	--	+	-	+	-
10	36	Yes/1980	Glu/20 doses	+	-	+	-
11	12	No	--	+	-	+	-
12	50	Yes/1988	Glu/60 doses	+	-	+	-
13	0.25	Yes/2004	Glu/60 doses	+	-	+	+
14	36	Yes/2017	Glu/60 doses	+	-	-	+

Glu = Glucantime^®^; Anfo = amphotericin.

## Data Availability

Data is available upon request from the authors; the data that support the findings of this study are available from the corresponding author upon reasonable request.

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
