# Peer review of "Overcoming the Negligence in Laboratory Diagnosis of Mucosal Leishmaniasis"

_pathogens, 2021, doi:10.3390/pathogens10091116_

Round 1

Reviewer 1 Report

The manuscript „Overcoming the negligence in laboratory diagnosis of mucosal leishmaniasis” described study conducted on mucosal leishmaniasis in northern region of Brazil.

Furthermore, the authors compared parasitological tests: direct parasitological examination by microscopy, conventional PCR targeting both kDNA and hsp70 and also qPCR for blood samples.

In my opinion the article provide important knowledge in the field of epidemiology. Futhermore, the authors suggested the need for large-scale molecular testing which is very important nowadays.

The manuscript is well written, however, I have few minor things for the authors:

  1. Please add numbers to the lines and paragraphs.
  2. Please consider to change „Examination of samples” instead of „Parasitological tests” in the M&M.
  3. Please read the whole manuscript once again and add spaces for example between „technical duplicates” in the paragraph „Real-time PCR for blood mucosal samples” (M&M).

Reviewer 2 Report

The manuscript focuses on the chronic parasitological diseases caused by the Leishmania species, for which laboratory diagnosis is mandatory. The authors evaluated more than 700 clinical samples from patients with cutaneous (CL) and mucosal leishmaniasis (ML), comparing the results of direct parasitological examination by microscopy (DP) and conventional PCR (cPCR). Finally they proposed a standardized method for collection and performing molecular diagnosis  from suspicious ML patients' samples for improving ML diagnosis. The article is well written and I recommand it for publication. However, some points have to be reconsider:

  1. Statistical analysis is missing and has to be added;
  2. Some references are so old. Please substitute them with novel one (doi: 10.3390/pharmaceutics13030343; 10.1007/s40121-020-00361-y; 10.1016/j.pt.2020.09.014; 10.1016/j.ijpharm.2021.120761).

Round 2

Reviewer 1 Report

Dear authors,

thank you for your response and corrections.

In my opinion the article is ready in this form for the to the publication in Pathogens.

Reviewer 2 Report

I recommended it for publication.